# Action-guided 3D Human Motion Prediction

**Jiangxin Sun**[*]
Sun Yat-sen University
sunjx5@mail2.sysu.edu.cn

**Zihang Lin**
Sun Yat-sen University
linzh59@mail2.sysu.edu.cn

**Xintong Han**
Huya Inc
hanxintong@huya.com

**Jian-Fang Hu**[†]
Sun Yat-sen University
hujf5@mail.sysu.edu.cn

**Jia Xu**
Huya Inc
xujia@huya.com

**Wei-Shi Zheng**
Sun Yat-sen University
wszheng@ieee.org

## Abstract

The ability of forecasting future human motion is important for human-machine interaction systems to understand human behaviors and make interaction. In this work, we focus on developing models to predict future human motion from past observed video frames. Motivated by the observation that human motion is closely related to the action being performed, we propose to explore action context to guide motion prediction. Specifically, we construct an action-specific memory bank to store representative motion dynamics for each action category, and design a query-read process to retrieve some motion dynamics from the memory bank. The retrieved dynamics are consistent with the action depicted in the observed video frames and serve as a strong prior knowledge to guide motion prediction. We further formulate an action constraint loss to ensure the global semantic consistency of the predicted motion. Extensive experiments demonstrate the effectiveness of the proposed approach, and we achieve state-of-the-art performance on 3D human motion prediction.

## 1 Introduction

Imagining and predicting the future motion of a person doing some actions (e.g., walking, standing up) is a natural ability of human beings. However, it is still very challenging for machines to achieve reasonable forecasting. In this work, we focus on 3D human motion prediction task which aims to forecast the future state of the 3D human body conditioned on several observed past video frames. Future motion prediction is important for many real-world applications like human-machine interaction and autonomous driving. The main challenge is that the future motion can vary greatly (i.e., the prediction space is large), which would increase the uncertainty and ambiguity of future motion prediction.

We observe that there exists a strong correlation between human motion and action category, which could be exploited to reduce such uncertainty and ambiguity. As illustrated in Figure 1, human action always contains some representative sub-processes such as swinging arm, throwing something out, etc. We name these sub-processes as *motion dynamics*. By exactly examining the videos in Figure 1, we further find that the motion dynamics could be shared across different subjects performing the same action. Motivated by the above observations, we propose to exploit representative motion dynamics for each action class to guide motion prediction and reduce the learning difficulty.

---

[*]This work was partially done when Jiangxin Sun was an intern at Huya Inc.

[†]Jian-Fang Hu is the corresponding author. Hu is also with Guangdong Province Key Laboratory of Information Security Technology, Guangzhou, China and Key Laboratory of Machine Intelligence and Advanced Computing, Ministry of Education, China.

35th Conference on Neural Information Processing Systems (NeurIPS 2021).

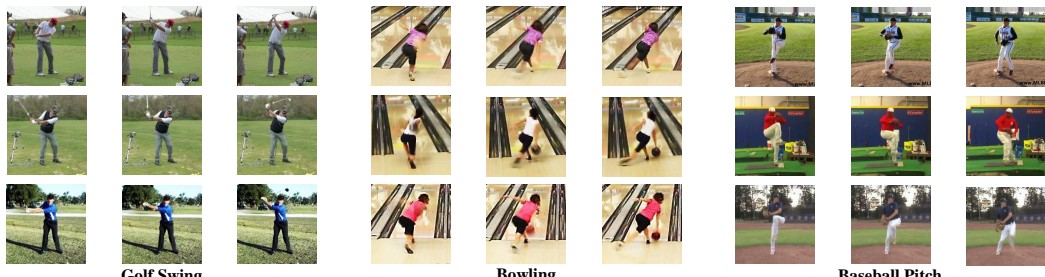

**Golf Swing**            **Bowling**            **Baseball Pitch**

Figure 1: Illustration of representative motion dynamics. This figure shows that the motion dynamics could be shared across different subjects who are performing the same action.

Specifically, we propose a novel framework to perform future 3D motion prediction. In this framework, we construct an action-specific memory bank to store the representative motion dynamics for each action class. The constructed memory bank stores both short-term and long-term motion dynamics to capture motions with different velocities. And based on the memory bank, we further design a query-read process to query past observed motion and read candidate future motion dynamics from the memory bank. The motion dynamics read from the memory bank are used to explicitly guide future motion prediction. To ensure global motion consistency, we further formulate an action constraint loss to keep the semantic consistency between the predicted motion and the observed past motion. The proposed approach achieves promising prediction results and outperforms previous methods by a considerable margin, which demonstrates the effectiveness of using action-specific motion dynamics to guide motion prediction.

Overall, our main contributions are: 1) we construct an action-specific memory bank to exploit the action-specific representative motion dynamics; 2) we propose a novel framework that can effectively utilize the action-specific dynamics stored in the memory bank for guiding motion prediction; 3) our approach achieves state-of-the-art performance on 3D human motion prediction.

## 2 Related work

**3D human motion prediction.**    3D human motion prediction has attracted long-standing interest in the community [5, 32, 41]. 3D human motion is typically modeled with 3D poses [23, 10, 25, 24, 28] or parametric 3D body models [1, 11, 16, 31, 17]. Earlier methods mainly perform 3D human motion prediction using techniques such as Hidden Markov Models [4], linear dynamical systems [29] and Gaussian process latent variable models [32, 33]. Recently, several deep learning-based approaches (e.g. methods employing recurrent neural networks or LSTMs [13]) are proposed to predict human poses from past 3D human skeletons [6, 45, 9, 41, 2, 7]. Since it is difficult to obtain in-the-wild 3D annotations, Zhang et al. [42] further proposed an autoregressive model named PHD to forecast future 3D human meshes from past 2D video inputs. However, it ignores the action information depicted in the observed video frames, which can serve as a strong prior to facilitate the prediction. In contrast, we aim to explore action context for motion prediction by designing an action-specific memory bank to discover the representative motion dynamics within each action class. With a query-read process, we explicitly utilize the information stored in the memory bank of each action class for guiding motion prediction.

**Memory networks.**    Internal memory has been widely used to capture long-term temporal dependencies within sequences (e.g., LSTMs [13], GRUs [8]) and can be implicitly updated in a recurrent process. In order to address the inherent instability of internal memory over long timescales, an external memory component has been developed to augment the sequence modeling and achieved improved performance on related tasks [34, 30, 18, 26]. The memory component can be read or written with an attention-based procedure and it can store useful information depicted in a sequence or a set of sequences. Recently, memory-based approaches have been developed in many vision tasks such as image captioning [37, 40], tracking [46, 19], and video understanding [35, 27]. Unlike traditional memory networks using a single memory to store all the information, we construct a memory bank to store independent memory components for each action category.

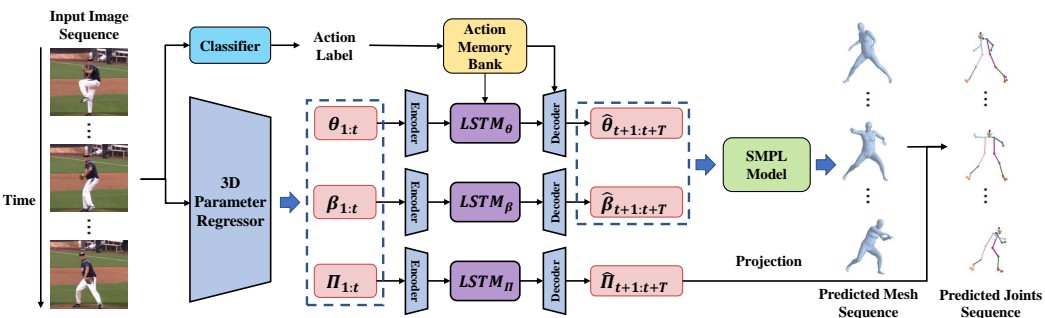

Figure 2: **An overview of the proposed framework.** We first extract 3D human body parameters from input videos with a 3D Parameter Regressor. Then we develop an encoder-LSTM-decoder architecture to perform prediction from these parameters. In order to utilize action context to improve the prediction, we further design an action memory bank to guide the prediction of the action-related parameters (i.e., the predicted SMPL pose parameters $\hat{\boldsymbol{\theta}}$).

## 3 Our approach

The goal of 3D human motion prediction is to forecast the future state of 3D human body presented in a given video. We follow [16, 42] to decompose the representation of 3D human body into three independent components $\boldsymbol{\theta}, \boldsymbol{\beta}$, and $\boldsymbol{\Pi}$. $\boldsymbol{\theta} \in \mathbb{R}^{72}$ and $\boldsymbol{\beta} \in \mathbb{R}^{10}$ denote the pose and shape parameters of SMPL model [21], respectively. The pose parameters $\boldsymbol{\theta}$ represent the global body rotation and the relative rotation of 23 body joints in axis-angle, which is closely related to action dynamics. The shape parameters $\boldsymbol{\beta}$ are the first 10 coefficients of a PCA space modeling body shapes, thus containing personal information such as height or waistline. Based on the constructed $\boldsymbol{\theta}$ and $\boldsymbol{\beta}$, the SMPL model generates a triangulated mesh $\mathcal{M}(\boldsymbol{\theta}, \boldsymbol{\beta}) \in \mathbb{R}^{N \times 3}$ with $N = 6890$ vertices. With a pre-trained linear regressor, the 3D coordinates of body joints $\mathbf{X}$ can be calculated from $\mathcal{M}(\boldsymbol{\theta}, \boldsymbol{\beta})$. $\boldsymbol{\Pi} = [s, t_x, t_y]$ is the weak-perspective camera parameter encoding the scaling, $x$-offset and $y$-offset information of an orthographic projection, which is used to infer the 2D coordinates of body joints $\mathbf{x}$ from 3D coordinates $\mathbf{X}$.

We address the problem of 3D human motion prediction by learning mappings $\Phi, \Psi$, and $\Theta$ to predict the 3D human body representations $\boldsymbol{\theta}_{t+1:T}, \boldsymbol{\beta}_{t+1:T}$ and $\boldsymbol{\Pi}_{t+1:T}$ of future un-observed video frames, respectively, from the representations $\boldsymbol{\theta}_{1:t}, \boldsymbol{\beta}_{1:t}$ and $\boldsymbol{\Pi}_{1:t}$ of the observed past frames. The prediction process can be mathematically expressed as:

$$\boldsymbol{\theta}_{t+1:T} = \Phi(\boldsymbol{\theta}_{1:t}), \boldsymbol{\beta}_{t+1:T} = \Psi(\boldsymbol{\beta}_{1:t}), \boldsymbol{\Pi}_{t+1:T} = \Theta(\boldsymbol{\Pi}_{1:t}). \tag{1}$$

The overall framework is presented in Figure 2. We follow the common practice in future prediction tasks [45, 9, 2, 7] to develop an encoder-LSTM-decoder architecture to perform the prediction. Specifically, we first employ an encoder to extend the representations $\boldsymbol{\theta}, \boldsymbol{\beta}, \boldsymbol{\Pi}$ to higher dimensional latent codes $\mathbf{B}^{(\theta)}, \mathbf{B}^{(\beta)}, \mathbf{B}^{(\Pi)}$, respectively. Then, three independent LSTMs are used to predict the latent representations for future frames. Finally, the decoder is employed to transform the predicted representations back to the space of $\boldsymbol{\theta}, \boldsymbol{\beta}, \boldsymbol{\Pi}$, which are fed into the SMPL model to construct 3D mesh/skeleton presentations.

Since the human motion has a strong correlation with the corresponding action category (e.g., one could better infer the future body motion when given the performing action), we seek to explore the action context with a novel action memory bank to guide the motion prediction in this work. The action memory bank is only used to help the prediction of pose parameters $\boldsymbol{\theta}$ as they are closely related to the action category, while for the prediction of action-irrelevant representations like shape parameters $\boldsymbol{\beta}$ and camera parameters $\boldsymbol{\Pi}$, we employ the standard encoder-LSTM-decoder architecture. In the following, we will mainly describe details of our framework for predicting SMPL pose parameters (i.e., learning mapping $\Phi$ in Eq. 1) with our proposed action memory bank.

### 3.1 Action-guided feature prediction

Actions of the same category always contain specific sub-processes and share some common dynamics (see Figure 1), which are useful to guide future motion prediction. In this work, in order to take

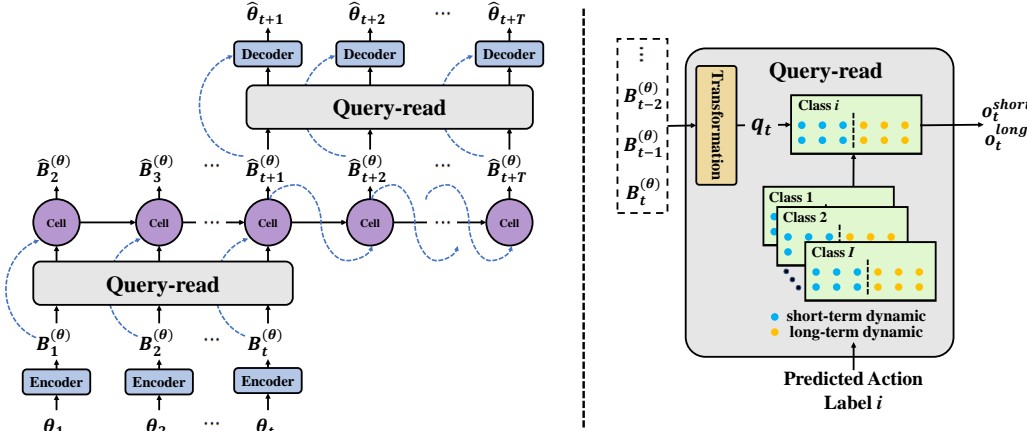

Figure 3: The architecture of the proposed action-guided predictor (left). We develop an action memory bank to store action-specific motion dynamics which are coupled with a query-read process (right) to guide motion prediction.

advantage of the action prior knowledge, we propose to construct a class-specific action memory bank to store the common dynamics of each action class and define a query-read process to utilize the dynamics stored in the memory bank as illustrated in Figure 3. In the query process, we calculate the similarity between the observed data and the dynamics contained in the pre-constructed memory bank conditioned on a predicted action class. In the read process, we combine the motion dynamics from the memory bank using the similarity as a soft weight. And the combined motion dynamic is finally used to guide motion prediction.

### 3.1.1 Action memory bank construction

Our action memory bank is constructed to store the representative motion dynamics appeared in each action class. Specifically, we represent the short-term motion dynamic as $\mathbf{s}_{t:t+\Delta t-1} = [\boldsymbol{\theta}_t, \boldsymbol{\theta}_{t+1}, \boldsymbol{\theta}_{t+2}, ..., \boldsymbol{\theta}_{t+\Delta t-1}]$, which consists of the initial pose $\boldsymbol{\theta}_t$ and the pose dynamics of the following $\Delta t - 1$ frames. That is to say, the proposed representation describes motion as the dynamics of human body joint rotations. In order to further capture long-term motion dynamics, we extend the sampling time step to $K$ and represent the long-term motion dynamic as $\mathbf{l}_{t:t+\Delta t-1} = [\boldsymbol{\theta}_t, \boldsymbol{\theta}_{t+K \times 1}, \boldsymbol{\theta}_{t+K \times 2}, ..., \boldsymbol{\theta}_{t+K \times (\Delta t-1)}]$. Then for each action class, we utilize the Affinity Propagation (AP) Clustering algorithm to cluster the short-term and long-term motions over all the training videos. We use L1 distance to measure the similarity of different motion patterns. We discard the clusters containing too few elements (less than 1%) and regard the remaining cluster centers as the representative motion dynamics of the corresponding action class to form our action memory bank $\mathcal{M}$ :

$$\mathcal{M} = \{(\mathbf{S}_i, \mathbf{L}_i) \mid i = 1, 2, ..., I\}, \tag{2}$$

where $\mathbf{S}_i \in \mathbb{R}^{(\Delta t \times 72) \times N_i^{short}}, \mathbf{L}_i \in \mathbb{R}^{(\Delta t \times 72) \times N_i^{long}}$ correspond to short-term motion and long-term motion memory for the action category $i$, respectively. Each item in the memory bank is a clustering center and $\Delta t$ is the time length of each item. $N_i^{short}$ and $N_i^{long}$ are the number of short-term and long-term items for action $i$ in the memory bank, respectively. $I$ denotes the number of action classes.

### 3.1.2 Motion feature prediction

With the constructed action memory bank, we take advantage of action-specific representative motion dynamics to guide the prediction. Specifically, we design a query-read process, which queries the past motion and then read the corresponding future motion dynamics from the memory bank. Inspired by previous memory based methods [30, 36, 38, 7], the query-read process is designed based on an attention mechanism. First, we employ an action recognition network [22] on the observed frames to predict the probability of action classes and take the class $i$ with the maximum classification confidence. Then, we compute the key matrices $\mathbf{K}_i^{short} \in \mathbb{R}^{C_k \times N_i^{short}}$ as $\mathbf{K}_i^{short} = \mathbf{W}_k \mathbf{S}_i^f$ and value matrices $\mathbf{V}_i^{short} \in \mathbb{R}^{C_v \times N_i^{short}}$ as $\mathbf{V}_i^{short} = \mathbf{W}_v \mathbf{S}_i^l$, where $\mathbf{S}_i^f, \mathbf{S}_i^l \in \mathbb{R}^{(\frac{\Delta t}{2} \times 72) \times N_i^{short}}$

represent the former and latter half motion in the memory bank, respectively. $\mathbf{W}_k \in \mathbb{R}^{C_k \times (\frac{\Delta t}{2} \times 72)}$ and $\mathbf{W}_v \in \mathbb{R}^{C_v \times (\frac{\Delta t}{2} \times 72)}$ are two learnable matrices. And for each time step $t$, we compute the query vector $\mathbf{q}_t \in \mathbb{R}^{C_k}$ from the encoded motion feature $\mathbf{B}_t^{(\theta)} \in \mathbb{R}^{C_B}$ by $\mathbf{q}_t = \mathbf{W}_q \mathbf{B}_{(t-\frac{\Delta t}{2}, t]}^{(\theta)}$, where $\mathbf{B}_{(t_1, t_2]}^{(\theta)} \in \mathbb{R}^{C_B \times (t_2 - t_1)}$ is the concatenation $[\mathbf{B}_{t_1+1}^{(\theta)}, \mathbf{B}_{t_1+2}^{(\theta)}, ..., \mathbf{B}_{t_2}^{(\theta)}]$. After that, in the query process, we compute the similarity between the query vector $\mathbf{q}_t$ and the key vectors in $\mathbf{K}_i^{short}$ as:

$$sim(\mathbf{K}_i^{short}, \mathbf{q}_t) = (\mathbf{K}_i^{short})^T \mathbf{q}_t. \tag{3}$$

In the read process, we read the future short-term motion dynamics $\mathbf{o}_t^{short} \in \mathbb{R}^{C_v}$ from the memory bank using the similarity as a soft weight, which can be formulated as:

$$\mathbf{o}_t^{short} = \mathbf{V}_i^{short} softmax(sim(\mathbf{K}_i^{short}, \mathbf{q}_t)). \tag{4}$$

Similarly, we can get the future long-term motion dynamics $\mathbf{o}_t^{long}$ with the memory bank by replacing $\mathbf{S}_i$ with $\mathbf{L}_i$. Finally, we concatenate the initial latent feature $\mathbf{B}_t^{(\theta)}$, $\mathbf{o}_t^{short}$ and $\mathbf{o}_t^{long}$ as the input to the LSTM to predict the latent features $\hat{\mathbf{B}}_{t+1:t+T}^{(\theta)}$ of future un-observed video frames.

### 3.1.3 Motion feature decoding

Once we get the predicted future latent feature $\hat{\mathbf{B}}_{t+1:t+T}^{(\theta)}$ for future frames $t+1:t+T$, the pose parameters $\hat{\theta}_{t+1:t+T}$ can be obtained by a decoding process. Similarly, we define our motion feature decoding process based on the query-read process. The main difference is that the employed key matrices $\mathbf{K}_i^{short,dec} = \mathbf{W}_k^{dec}\mathbf{S}_i$ and value matrices $\mathbf{V}_i^{short,dec} = \mathbf{W}_v^{dec}\mathbf{S}_i$ are computed based on the complete motion $\mathbf{S}_i \in \mathbb{R}^{(\Delta t \times 72) \times N_i^{short}}$ rather than partial motion $\mathbf{S}_i^f, \mathbf{S}_i^l \in \mathbb{R}^{(\frac{\Delta t}{2} \times 72) \times N_i^{short}}$ in the memory bank, as the full motion features are available in the decoding stage. The query vector is formulated as $\mathbf{q}_t^{dec} = \mathbf{W}_q^{dec}\hat{\mathbf{B}}_{(t-\Delta t, t]}^{(\theta)}$. Thus, short-term motion dynamics $\mathbf{o}_t^{short,dec}$ read from the memory bank can be formulated as:

$$\mathbf{o}_t^{short,dec} = \mathbf{V}_i^{short,dec} softmax((\mathbf{K}_i^{short,dec})^T \mathbf{q}_t^{dec}). \tag{5}$$

The long-term motion dynamics $\mathbf{o}_t^{long,dec}$ can be obtained by replacing $\mathbf{S}_i$ with $\mathbf{L}_i$. Then the predicted latent features $\hat{\mathbf{B}}_t^{(\theta)}$, $\mathbf{o}_t^{short,dec}$ and $\mathbf{o}_t^{long,dec}$ are concatenated together as the input of the decoder, generating 3D pose parameters $\hat{\theta}_{t+1:t+T} = [\hat{\theta}_{t+1}, \hat{\theta}_{t+2}, ..., \hat{\theta}_{t+T}]$.

### 3.2 Model training

Our objective in future motion prediction is to learn the model parameters such that the gap between the predicted motion and the ground-truth motion is minimized. To achieve this, we define the prediction loss $\mathcal{L}_{pred}$ as a series of MSE losses formulated as follows:

$$\mathcal{L}_{pred} = \mathcal{L}_{parameter} + \mathcal{L}_{latent} + \mathcal{L}_{joint}, \tag{6}$$

$$\mathcal{L}_{parameter} = \sum_{t=2}^{T} \left\| \theta_t - \hat{\theta}_t \right\|_2^2 + \left\| \beta_t - \hat{\beta}_t \right\|_2^2 + \left\| \mathbf{\Pi}_t - \hat{\mathbf{\Pi}}_t \right\|_2^2, \tag{7}$$

$$\mathcal{L}_{latent} = \sum_{t=2}^{T} \left\| \mathbf{B}_t^{(\theta)} - \hat{\mathbf{B}}_t^{(\theta)} \right\|_2^2 + \left\| \mathbf{B}_t^{(\beta)} - \hat{\mathbf{B}}_t^{(\beta)} \right\|_2^2 + \left\| \mathbf{B}_t^{(\Pi)} - \hat{\mathbf{B}}_t^{(\Pi)} \right\|_2^2, \tag{8}$$

$$\mathcal{L}_{joint} = \sum_{t=2}^{T} \left\| \mathbf{X}_t - \hat{\mathbf{X}}_t \right\|_2^2 + \left\| \mathbf{v}_t \odot (\mathbf{x}_t - \hat{\mathbf{x}}_t) \right\|_2^2. \tag{9}$$

Here, $\mathcal{L}_{parameter}, \mathcal{L}_{latent}$ are the MSE losses of the body parameters $\theta, \beta, \mathbf{\Pi}$ and their corresponding latent feature $\mathbf{B}^{(\theta)}, \mathbf{B}^{(\beta)}, \mathbf{B}^{(\Pi)}$, respectively. $\mathcal{L}_{joint}$ is the MSE loss for 3D keypoints and 2D keypoints. $\mathbf{v}_t$ is the visibility indicator over ground truth 2D keypoints.

Considering the importance of prior knowledge, we follow [42, 15] to employ a prior loss $\mathcal{L}_{prior}$, which is a commonly used constraint to ensure the reasonability of the predicted human body according to human prior. It includes human pose reasonability prior loss [15, 16], shape regularization prior loss[3] and shape consistency loss [16].

In order to keep the semantic consistency between predicted motion and ground-truth motion, we further propose an action constraint loss $\mathcal{L}_{action}$. We train a 3D skeleton-based action classification network $\phi$ [20] on the training videos, and force the action classification results $\hat{\mathbf{C}} = \phi(\hat{\mathbf{X}})$ using the predicted 3D joints $\hat{\mathbf{X}}$ to be similar to the results $\mathbf{C} = \phi(\mathbf{X})$ obtained with ground-truth 3D joints $\mathbf{X}$. Here, we calculate a Kullback-Leibler divergence loss between $\hat{\mathbf{C}}$ and $\mathbf{C}$ to constraint motion prediction as $\mathcal{L}_{action} = KL(\hat{\mathbf{C}}, \mathbf{C})$. Thus, the complete loss formulation is:

$$\mathcal{L} = \mathcal{L}_{pred} + \mathcal{L}_{prior} + \mathcal{L}_{action}. \tag{10}$$

Since directly training the whole encoder-LSTM-decoder architecture with randomly initialized parameters can be unstable, we introduce a two-stage optimization approach. In the first stage, we remove all the LSTMs in the framework and only train the encoders and decoders by minimizing the body parameter loss $\mathcal{L}_{parameter} = \left\| \boldsymbol{\theta} - \hat{\boldsymbol{\theta}} \right\|_2^2 + \left\| \boldsymbol{\beta} - \hat{\boldsymbol{\beta}} \right\|_2^2 + \left\| \boldsymbol{\Pi} - \hat{\boldsymbol{\Pi}} \right\|_2^2$. This achieves a good initialization of the encoder and decoder parameters. In the second stage, we train the whole framework to optimize 3D human motion prediction with loss function $\mathcal{L}$ defined in Eq. (10).

## 4 Experiments

**Datasets.**  We evaluate the proposed method on Human3.6M [14] and Penn Action [44] datasets, which contain human action videos captured under different views with various motions. Human3.6M [14] contains 3.6 million 3D human poses and corresponding images. To construct this set, 11 professional actors are invited to perform 17 activities in an indoor laboratory: discussion, smoking, taking photos, etc. This set is well annotated with 3D joint coordinates, joint angles, person bounding boxes, and 3D laser scans of each actor. Following [42], we use the data of Subjects 1, 6, 7, and 8 as the training set, data of Subject 5 as the validation set, and data of Subjects 9 and 11 as the test set. The Penn Action [44] dataset consists of 2326 in-the-wild sport videos involving 15 actions such as baseball pitch, bowling, golf swing, etc. Since all the videos are collected in the wild, the ground truth 3D poses are not available in this set. The annotations only contain action class labels, 2D bounding boxes, and human body joints (2D locations and visibility). We use the methods in [42] to generate the corresponding pseudo 3D annotations and use them to train the model.

**Evaluation metrics.**  The same as that in [42], for the Human3.6M dataset with 3D human pose annotations, we report the PA-MPJPE (mean error of per joint position after applying Procrustes Alignment [12]) in *mm*. For the Penn Action dataset with 2D pose annotations, we measure the PCK (percentage of correct keypoints [39]) at $\alpha = 0.05$. Following [42], we input 15 past frames to the model and predict the motion of the next 30 frames. We start the prediction from every 25th frame for Human3.6M dataset and start from every frame for Penn Action dataset. For a fair comparison with PHD [42], we also provide the results of our approach using Dynamic Time Warping (DTW), which can reduce the prediction error caused by unreliable velocity prediction. The DTW intends to find an optimal matching between the ground truth and predicted future motion sequence after time-warping, so that the similarity between them is maximized. However, we would like to point out that using DTW for evaluation is unreasonable, since we have no chance of obtaining the ground truth to warp the prediction in real-world applications.

**Network architecture.**  We employ the 3D human reconstruction model in PHD [42] as our 3D human body parameter regressor to extract the decoupled 3D motion representations. It consists of a ResNet50 image encoder, a causal 1D temporal convolutional encoder and a regressor with iterative optimization. The dimensions of the encoded 3D human body parameters $\mathbf{B}^{(\theta)}, \mathbf{B}^{(\beta)}, \mathbf{B}^{(\Pi)}$ are 1024, 256, 64, respectively. Both the encoder and decoder are designed as 3 stacked FC-ReLU-BN blocks. The number of LSTM hidden units for predicting $\boldsymbol{\theta}$, $\boldsymbol{\beta}$, $\boldsymbol{\Pi}$ are 1024, 256, 64, respectively. The temporal length $\Delta t$ of each memory item is 8 and sampling time interval $K$ of long-term motion is set to 3. The channels of both key vector $C_k$ and value vector $C_v$ are kept same and set to 512.

**Learning details.**  We train our predictors using SGD algorithm with a Nesterov momentum of 0.9, where the 3D human body parameter regressor is frozen with the parameters pre-trained in PHD [42]. In the first training stage, we train the encoder and decoder with learning rate 0.01 for 6 epochs. In the second stage, we train the whole prediction block in an end-to-end manner for 30 epochs. The

Table 1: Comparison results with recent approaches for human motion prediction.

| | Method | Human 3.6M (PA-MPJPE ↓) | | | | | Penn Action (PCK ↑) | | | | |
|---|---|---|---|---|---|---|---|---|---|---|---|
| | | 1 | 5 | 10 | 20 | 30 | 1 | 5 | 10 | 20 | 30 |
| | Oracle | 56.9 | 56.9 | 56.9 | 56.9 | 56.9 | 81.6 | 81.6 | 81.6 | 81.6 | 81.6 |
| With DTW | Nearest Neighbor | 90.3 | 95.1 | 100.6 | 108.6 | 114.2 | 63.2 | 61.5 | 60.6 | 58.7 | 57.8 |
| | Constant | 59.7 | 65.3 | 72.8 | 84.3 | 90.4 | 78.3 | 71.7 | 64.9 | 56.2 | 49.7 |
| | PHD [42] | 57.7 | 59.5 | 61.1 | 62.1 | 65.1 | 81.2 | 80.0 | 79.0 | 78.2 | 77.2 |
| | Ours | **57.3** | **58.9** | **60.3** | **61.4** | **62.8** | **81.4** | **80.6** | **79.7** | **79.1** | **78.5** |
| Without DTW | Nearest Neighbor | 90.3 | 99.8 | 110.3 | 124.7 | 133.3 | 62.5 | 57.6 | 53.7 | 44.6 | 41.1 |
| | Constant | 59.7 | 71.4 | 85.9 | 101.4 | 102.8 | 78.3 | 65.5 | 54.6 | 42.3 | 32.7 |
| | PHD [42] | 57.7 | 61.2 | 64.4 | 67.1 | 81.1 | 81.2 | 77.2 | 72.4 | 67.9 | 60.1 |
| | Ours | **57.3** | **59.6** | **61.7** | **62.5** | **75.9** | **81.4** | **79.1** | **76.7** | **72.8** | **66.5** |

learning rate is initialized to 0.01 and decreased to 0.001 after 12 epochs. It took about 3 days for Human 3.6M dataset and 20 hours for Penn Action dataset to train our framework using $4\times$V100 GPUs. For training the 3D skeleton-based action classifier, we use the publicly available code of [20] with default parameters.

## 4.1 Experimental results

We compare our approach with the state-of-the-art approach PHD [42] and two other baselines (Nearest Neighbor and Constant as discussed in [42]). The Nearest Neighbor baseline takes the input conditioning frames to find the closest sequence in the training set based on the Euclidean distance of 3D joint locations, and regards the motion reconstructed from the following frames as the prediction result. The Constant baseline regresses the 3D human body from the last observed frame and regards it as a constant prediction result for future frames. In addition, we also implement an Oracle approach by directly reconstructing 3D human motion using the 3D parameter regressor in our model from the corresponding ground-truth frames, which can be regarded as an upper bound for our prediction.

We report our experimental results on the Human3.6M [14] dataset and the Penn Action [44] dataset in Table 1. For Human3.6M set, our method consistently outperforms all the competitors by a considerable margin for both five-frame prediction (0.6mm improvement with DTW and 1.6mm improvement without DTW) and thirty-frame prediction (3.7mm improvement with DTW and 5.6mm improvement without DTW). The results verify that predicting features with action-related prior knowledge can achieve better prediction results. The performance improvement gets larger as the length of prediction increases. This is as expected, as the developed action memory bank and action loss constraint can be used as strong guidance for motion prediction, which effectively narrow the prediction space. Thus, the prediction difficulty could be greatly reduced, especially for the prediction of long sequences. Our results on the Penn Action [44] dataset show that our method also achieves state-of-the-art performances for human motion prediction. Specifically, our method achieves 80.6% for five-frame prediction and 78.5% for thirty-frame prediction with DTW. Without DTW, our method can perform 79.1% for five-frame prediction and 66.5% for the thirty-frame prediction, which are 1.9% and 6.4% higher than the results reported in PHD [42]. The promising results show that the action context can provide rich information for the 3D motion prediction. We also note that the performance improvement obtained by our method using DTW is smaller than that of PHD. This is because that the developed action memory bank with both short-term and long-term dynamics has already captured motion velocity information and better resolved the velocity ambiguity.

In Figure 4, we present some visualization results for the predicted 3D human mesh. For the first example, we can observe that our predictions align much better to the ground truth than predictions of PHD [42]. It indicates that the developed action memory bank can help predict a reasonable motion velocity for the ongoing action sequences. For the second example, the poses predicted by our method are more accurate, which demonstrates the proposed method can learn the motion variation better.

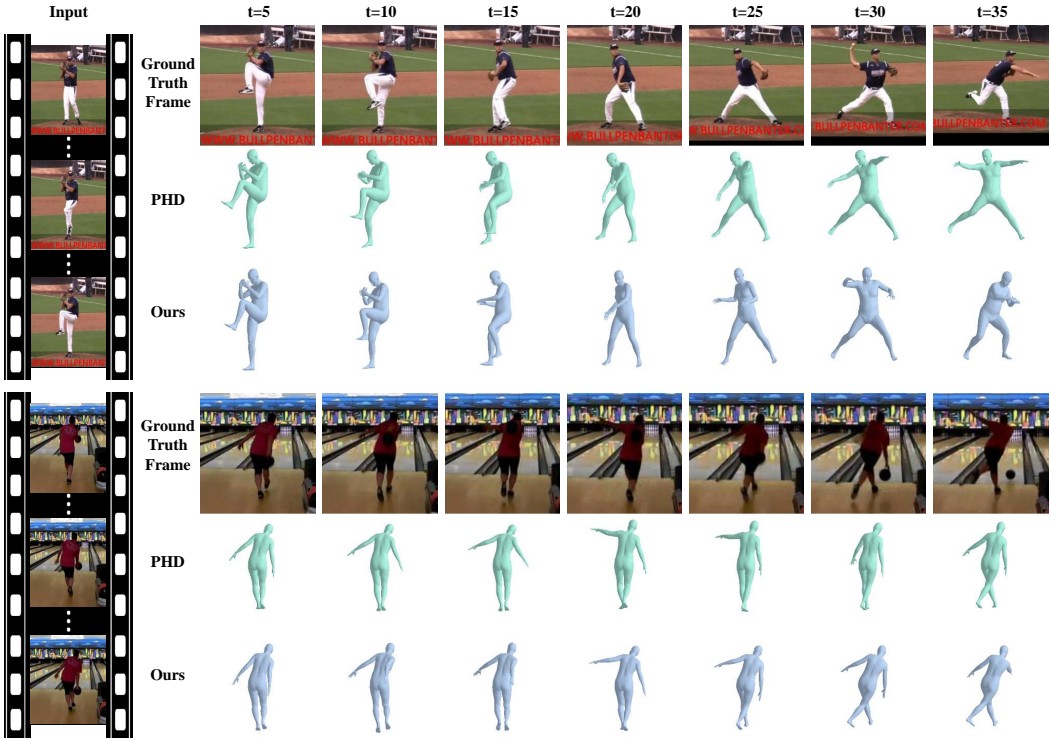

Figure 4: **Visualization of 3D human motion prediction.** From left to right: the observed input video and motion predicted at different time steps. We provide samples of baseball pitch and bowling. For each sample, the top row contains the ground-truth frames. The results obtained by PHD [42] and our approach are presented in the mid and bottom row, respectively.

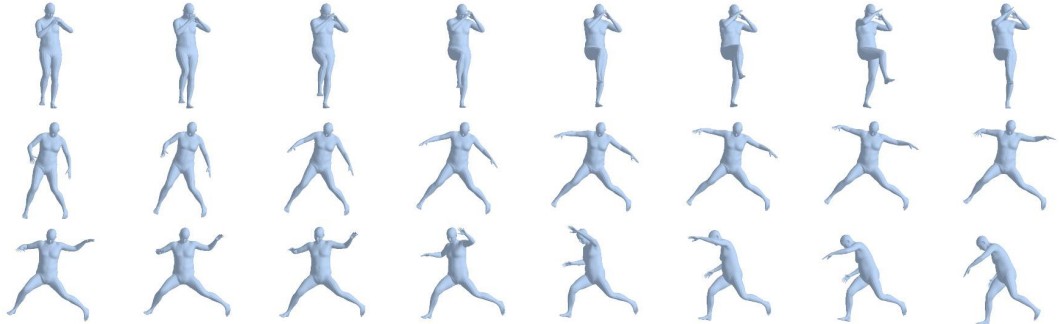

Figure 5: Each row visualizes an item in our action memory bank for the baseball pitch class.

In Figure 5, we provide some visualization results of the motion dynamics in our memory bank. As shown, our memory bank contains some representative motion dynamics for baseball pitch, which could provide a strong prior for motion prediction.

## 4.2 Ablation studies

In this section, we conduct extensive ablation experiments for future human motion prediction on the Human3.6M dataset [14] to study the influence of each component in our framework. We report the results without DTW, which is more reasonable than the model with DTW in real-world applications.

**Influence of action context for prediction.** We provide an in-depth analysis on the proposed motion prediction method by gradually adding the components and tabulated the results in Table 2. We start from a baseline that only contains the encoder, LSTM, and decoder. Then we query

Table 2: Evaluation of our action memory bank and other components on Human3.6M dataset.

| Method | Reconstruction error ↓ | | | |
| --- | --- | --- | --- | --- |
| | 5 | 10 | 20 | 30 |
| Baseline | 63.1 | 66.5 | 68.9 | 83.3 |
| + Prediction with Bank | 61.2 | 63.8 | 65.7 | 79.6 |
| + Decoding with Bank | 59.6 | 61.9 | 63.4 | 77.1 |
| + Action Constraint | **59.6** | **61.7** | **62.5** | **75.9** |

Table 3: Evaluation of action context modeling on Human3.6M dataset.

| Method | Reconstruction error ↓ | | | |
| --- | --- | --- | --- | --- |
| | 5 | 10 | 20 | 30 |
| Baseline | 63.1 | 66.5 | 68.9 | 83.3 |
| Action Label | 61.8 | 64.9 | 66.5 | 80.2 |
| Action Memory Bank | **59.6** | **61.7** | **62.5** | **75.9** |

Table 4: Evaluation of our action-specific memory bank on Human3.6M dataset

| Method | Reconstruction error ↓ | | | |
| --- | --- | --- | --- | --- |
| | 5 | 10 | 20 | 30 |
| Baseline | 63.1 | 66.5 | 68.9 | 83.3 |
| Action-agnostic Bank | 61.4 | 64.6 | 66.3 | 80.7 |
| Action-specific Bank | **59.6** | **61.9** | **63.4** | **77.1** |

Table 5: Evaluation of utilizing the long-term motion dynamics on Human3.6M dataset.

| Method | Reconstruction error ↓ | | | |
| --- | --- | --- | --- | --- |
| | 5 | 10 | 20 | 30 |
| Only Short | 60.1 | 62.4 | 63.7 | 77.3 |
| $K = 2$ | **59.4** | 61.9 | 63.1 | 76.8 |
| $K = 3$ | 59.6 | **61.7** | **62.5** | **75.9** |
| $K = 4$ | 60.0 | 62.1 | 63.0 | 76.4 |
| $K = 5$ | 60.1 | 62.2 | 63.5 | 77.0 |

Table 6: Influence of the predicted action label on Human3.6M dataset.

| Accuracy | Reconstruction error ↓ | | | |
| --- | --- | --- | --- | --- |
| | 5 | 10 | 20 | 30 |
| Ground-truth (100%) | 59.6 | 61.4 | 61.9 | 74.1 |
| Classifier (87%) | 59.6 | 61.7 | 62.5 | 75.9 |
| 80% | 59.9 | 62.0 | 62.8 | 76.3 |
| 75% | 60.1 | 62.2 | 63.4 | 76.9 |
| 70% | 60.7 | 62.6 | 64.1 | 78.2 |
| 65% | 61.1 | 63.4 | 65.2 | 79.7 |
| 60% | 61.8 | 64.2 | 67.0 | 81.5 |

and read motion dynamics from the memory bank and treat it as an additional input to the LSTM (termed as *+Prediction with Bank*). We find that it can bring 3.7mm improvement for the prediction of 30-frame prediction. Further, we use the motion dynamics captured in the memory bank to guide our decoding process (termed as *+Decoding with Bank*). The performance is improved by more than 1.6mm. Finally, we add the action constraint loss $\mathcal{L}_{action}$ to capture more global action context, which measures the consistency of the predicted motion sequence from a global perspective. We can observe more improvement in the prediction, especially for the prediction of longer frames (20 and 30 frames). The results demonstrate that the action context captured in the action memory bank and action constraint loss can serve as strong guidance for motion prediction. To further demonstrate the importance of modeling action context and the effectiveness of the proposed memory bank, we implement another simple one-hot baseline which models action context by taking the action label (i.e., a one-hot vector) indicating the action class as additional input to the motion predictor. As shown in Table 3, the one-hot baseline can also reduce the reconstruction error, indicating that injecting action context helps motion prediction. However, this one-hot baseline is much less accurate than our proposed approach, which further demonstrates the effectiveness of our proposed memory bank.

**Action-agnostic vs. action-specific memory bank.** In our action memory bank, we perform clustering for each action class separately (i.e., our memory bank is constructed in an action-specific manner). We can also construct an action-agnostic memory bank by clustering over all the dynamic samples without considering the action labels. The experimental results are presented in Table 4. For a fair comparison, we report our approach without the action constraint. As shown, our approach with the action-agnostic bank can already bring consistent improvement over the baseline. And our proposed action-specific memory bank further promotes the performance, which demonstrates the effectiveness of exploring the action-specific context for future motion prediction.

**Benefits of long-term dynamics in memory bank.** In our memory bank, short-term dynamic memory item and long-term dynamic memory item (with a sampling interval $K$) capture motion dynamics at different temporal scales. Here, we study the influence of $K$ as shown in Table 5. As can be seen, compared with the setting that only uses short-term motion dynamics, adding long-term motion dynamics can always improve the prediction performance, which indicates that the long-term

motion dynamic complements well with the short-term dynamics. The combination of them can provide more useful information to guide future motion prediction. Setting $K = 3$ achieves the best prediction performance among all the cases.

**Impact of the predicted action label.**    In the prediction process, we employ an action classifier to obtain the action label of the sequence, so that the predictor can query and read relevant dynamic information from the memory bank of the corresponding action. However, the action label can be wrongly recognized in some hard cases. To study the influence of the predicted action labels, we conduct experiments by replacing some low confidence predictions (but correctly predicted) with wrong labels. Our classifier can achieve an accuracy of 87% and we gradually reduce the accuracy to 80%, 75%, 70%, 65% and 60%. The detailed results are presented in Table 6. The motion prediction performance will increase when ground truth labels are provided. Meanwhile, the performance decreases as the recognition accuracy goes down. It is worth noting that when the accuracy is between 75% and 100%, the prediction performance is relatively stable with a difference less than 1mm in terms of 30-frame prediction, which demonstrates the robustness of the proposed method.

## 5    Conclusion

In this work, we propose to exploit the representative motion dynamics of each action class for 3D human motion prediction. Specifically, we construct an action-specific memory bank to store the representative motion dynamics. Based on the constructed memory bank, we formulate a query-read process to read motion dynamics from the memory bank guiding motion prediction. An action constraint loss is formulated to enhance the semantic consistency of the predicted 3D motion. Extensive experiments on two benchmarks demonstrate the effectiveness of using action-specific motion dynamics to guide motion prediction. For social impact, we believe that our model can be used to predict human motion for autonomous driving, human-computer interaction, and augmented/virtual reality applications. We also acknowledge that there exists potential negative social impact such as generating malicious fake videos conditioned on the predicted human motion with GANs.

**Limitations and Future Work.**    Our proposed method faces three main limitations: 1) our model relies heavily on the action classification, 2) we develop models to predict the motion features outputted by a fixed 3D parameter regressor without optimizing it, 3) we predict future 3D human motion for each appeared person independently without considering its interactions with the environment. In the future, we would consider learning the recovery and prediction of human shape and motion jointly in an end-to-end manner, as well as modeling human-human/human-object interactions as in [43]. Also, we intend to develop a more intricate design for the memory bank such as action hierarchy, temporal pyramid and part-level body motion.

## Acknowledgments and Disclosure of Funding

This work was supported partially by the NSFC (U1911401, U1811461, 62076260, 61772570), Guangdong Natural Science Funds Project (2020B1515120085, 2018B030312002), Guangdong NSF for Distinguished Young Scholar (2018B030306025), Guangzhou Research Project (201902010037), Research Projects of Zhejiang Lab (2019KD0AB03), and the Key-Area Research and Development Program of Guangzhou (202007030004).

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
