# Supplemental Material of
# Action-guided 3D Human Motion Prediction

## A    More experimental evaluation

In this section, we provide more experimental evaluation on our approach. In Figure 1, we present visualization results of the predicted 3D human mesh from Human3.6M dataset [1]. We can observe that our approach can better handle tiny cues of motion dynamics.

Table 1 and Table 2 provide additional ablation results for our approach on the Penn Action dataset [6] when DTW is not applied. Detailed setting of these experiments can be found in Section 4.2 of the main paper.

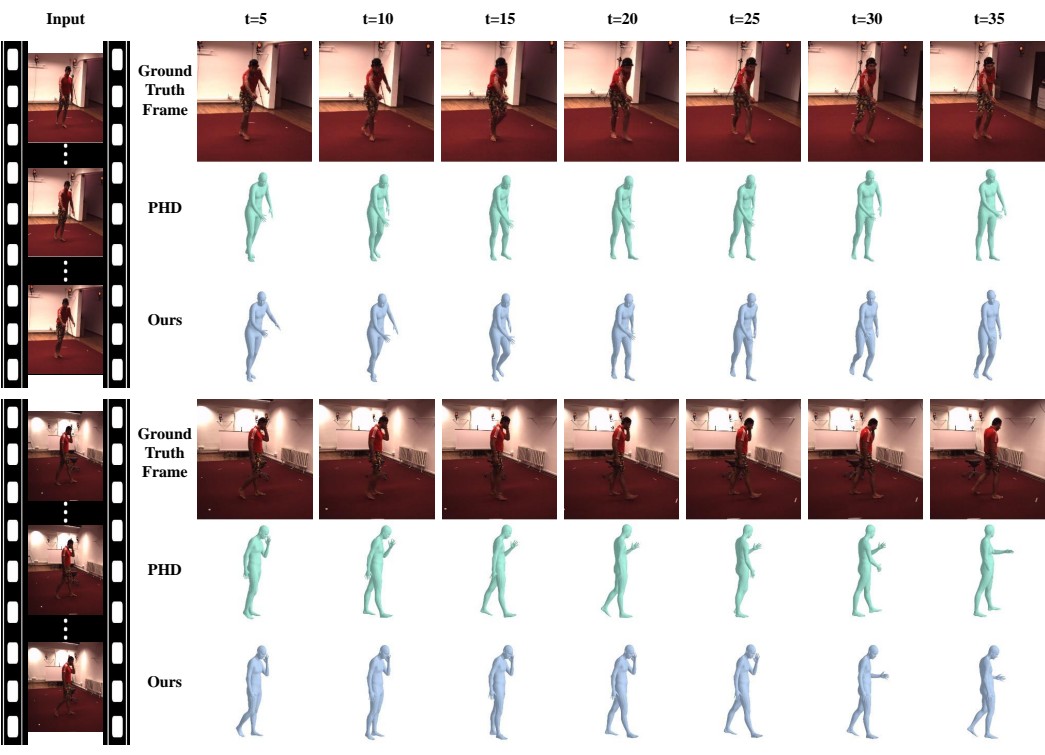

Figure 1: **Visualization of 3D human motion prediction.** From left to right: the observed input video and motion predicted at different time steps. We provide samples of walking dog and phoning. For each sample, the top row contains the ground-truth frames. The results obtained by PHD [5] and our approach are presented in the mid and bottom row, respectively.

## B    Licenses of referenced assets

We provide the links pointing to the licenses of our referenced assets, including pre-trained models and datasets.

**Human3.6M dataset [1]**    http://vision.imar.ro/human3.6m/eula.php

Table 1: Evaluation of our action context modeling on Penn Action dataset without DTW.

| Method | PCK ↑ | | | |
| --- | --- | --- | --- | --- |
| | 5 | 10 | 20 | 30 |
| Baseline | 76.8 | 71.5 | 66.8 | 58.6 |
| + Prediction with Bank | 77.9 | 74.2 | 69.5 | 62.1 |
| + Decoding with Bank | 78.6 | 75.9 | 71.4 | 64.7 |
| + Action Constraint | **79.1** | **76.7** | **72.8** | **66.5** |

Table 2: Evaluation of our action-specific memory bank on Penn Action dataset without DTW.

| | PCK ↑ | | | |
| --- | --- | --- | --- | --- |
| | 5 | 10 | 20 | 30 |
| Baseline | 76.8 | 71.5 | 66.8 | 58.6 |
| Action-agnostic Bank | 77.5 | 73.4 | 68.6 | 61.3 |
| Action-specific Bank | **78.6** | **75.9** | **71.4** | **64.7** |

**Penn action dataset [6]**    http://dreamdragon.github.io/PennAction/

**PHD model [5]**    https://github.com/jasonyzhang/phd/blob/master/LICENSE

**SMPL model [3]**    https://smpl.is.tue.mpg.de/modellicense

**RGB-based classifier [4]**    https://github.com/dluvizon/deephar/blob/master/LICENSE.md

**Skeleton-based classifier [2]**    https://github.com/kenziyuliu/MS-G3D/blob/master/LICENSE