# OpenReview forum: "Action-guided 3D Human Motion Prediction"
_NeurIPS.cc/2021/Conference — NeurIPS 2021 Poster_

### Official Review · Reviewer_CSBT · 2021-07-14

**Rating:** 6
**Confidence:** 4

**Summary:**

This paper introduces a new pipeline for human motion prediction with the help of an action dynamics memory bank. Based on the previous work PHD, the stored action motion dynamics from the training data clustering and action classification task can guide the prediction better. In experiments, the proposed method outperforms its counterpart and several baselines on two widely-used benchmarks.

**Ethical Concerns:**

No.

**Limitations And Societal Impact:**

Yes.

**Main Review:**

originality:
The main framework is following the PHD. The main novelties come from the action dynamics memory and proxy action classification.

quality:
Sound design and good results.

clarity:
Well written and easy to follow.

significance:
Important field, good improvements.

Detailed comments

Pros:
+ Sound design of the action dynamics memory, many actions indeed have some common primitives.

+ Good results compared to the PHD.

+ It is interesting to apply the memory module to motion prediction.

Cons:
- Relying heavily on the action classification. A coin always has two sides. This is one of my main concerns.

1, we must classify all actions. Unseen ones are not considered, while the other methods do not have this concern. Moreover, when we should process many actions, scaling the memory bank would be tricky.

2, in use, how to decide class-i? One-hot vector via thresholding or soft probability? It looks that the authors used the latter one. It should be clarified.

3, a person can perform many different actions simultaneously. How to handle this very common situation? For example, eating and talking.

4, did not consider the action hierarchy. In clustering, should we consider the inclusion relations between actions? It looks very vital to me, e.g., the atomic dynamics of bowing and holding-sth. Though we can ignore this and directly see that all actions are mutually exclusive, then the rationality of motion prediction seems robust for different actions. In addition, many actions usually share similar atomic motion dynamics. Why not consider this phenomenon in the memory module design?

5, the atomic dynamics claim is the critical insight to support the whole work, but just several visualizations are shown. I suggest the authors provide more analysis to prove the atomic and common properties of the found dynamics in the memory module. And how the query works in some ideal situations, can the dynamics in memory well match the different cases.

6, the possible bias from the clustering. The long-tailed distribution is common in action datasets. How this affects the clustering and representative dynamics finding?

7, usually, many part-level actions are the essential elements in actions instead of the whole-body motions. Can finer-grained dynamics further improve the prediction?

- The proposed method is based on the PHD. The novel parts are the memory module and action classification task. Though I think this would not hurt much, more discussions about the difference between the two methods are essential.

- Using the pre-trained regressor from PHD. Is this fair for the comparison between the two methods? How about training from scratch? Would the improvement be marginal? As the main competitor is the PHD, this would be very important. BTW, the author has honestly mentioned this in the limitation discussion.

- The memory only works for the pose prediction. The pose and shape are highly correlated. Can we omit this or leave it to the shared backbone or SMPL? An ablation study should be supplemented.

- v_t: how to get the visibility indicator? Need clarification.

- w/ and w/o DTW, the performance improvements differ significantly. Please analyze this phenomenon.

- Fig 4: maybe some indications like red boxes on the unreasonable predicted human parts would highlight the comparison more clear. The current figure is complicated for me to find the difference and how the proposed method outperforms PHD.

Overall, this is an interesting work to utilize more prior knowledge from motion data to advance motion prediction. But many clarifications and discussions are missing.
I will decide the final rating according to the response from the authors.

------Post-rereview
Thanks for the response from the authors.
After reading all reviews and responses, my most concerns were addressed.
I keep the rating and positive to this wors. For more please refer to the below response.

**Time Spent Reviewing:**

8

---

> ### Author Response · Authors · 2021-08-10
> **Response To Reviewer CSBT (First Part)**
>
> **Q1: Relying heavily on the action classification.**
>
> Re: Indeed, we have conducted experiments without action classification and provide detailed results in Table 3 (Action-agnostic Bank). As shown, the action-agnostic bank, which clusters atomic motion dynamics in a purely unsupervised way without using the action label, can also bring some improvements (about 2.6mm for thirty-frame prediction). However, the action-agnostic bank could miss some important motion dynamics, which are not representative for most of the actions but crucial for a certain action due to the long-tailed distribution (refer to Q8 for more discussions). In contrast, clustering over each action class makes it easier to explore such motions and thus improves the performance. Therefore, in the proposed method, we utilize an action classification network to infer the class label. Since our main purpose is to reveal that action information can effectively guide the motion prediction, we did not address the common issues in action classification at the same time within the limited pages. We leave them as future works to improve our newly proposed framework, e.g. fix the classification issues or further exploit atomic dynamics across all action classes to reduce the reliance on action classification. And we believe that our work can facilitate future research on human motion prediction with action context.
>
> **Q2: We must classify all actions. Unseen ones are not considered, while the other methods do not have this concern. Moreover, when we should process many actions, scaling the memory bank would be tricky.**
>
> Re: In the submission, a standard action classification framework is employed to query representative motion dynamics of each specific action class. Our goal is to develop models to explore action context for guiding our motion prediction. The mentioned unseen action and class scaling problems are indeed two challenging research directions in the community, which require to be specifically studied in-depth and are out of the scope of this work. In the future, we will consider addressing these problems so that our approach can easily scale to more and even unseen action classes.
>
> **Q3: In use, how to decide class-i?**
>
> Re: We take the class with the maximum classification confidence. We will clarify it.
>
> **Q4: A person can perform many different actions simultaneously. How to handle this very common situation?**
>
> Re: A straightforward way to handle this situation is to perform multi-label classification rather than single-label classification as done in our current framework, and combine the information from the memory banks of multiple actions to guide motion prediction. We believe this can model multiple simultaneous actions to some extend.
>
> **Q5: did not consider the action hierarchy. In clustering, should we consider the inclusion relations between actions? It looks very vital to me, e.g., the atomic dynamics of bowing and holding-sth. Though we can ignore this and directly see that all actions are mutually exclusive, then the rationality of motion prediction seems robust for different actions.**
>
> Re: Considering the action hierarchy and the inclusion relations between actions should be helpful (e.g., design cross-action or hierarchical memory banks). However, how to define and develop the action hierarchy for clustering itself is complicated and remains a challenging problem ([1][2][3]). We have not fully explored this aspect and leave it as future work.
>
> **Q6: Many actions usually share similar atomic motion dynamics. Why not consider this phenomenon in the memory module design?**
>
> Re: Thank you for your comment. Mining shared atomic motion dynamics among different actions is a promising idea. However, it could be difficult to mine useful shared atomic motions across each pair of action classes ([2][4][5]), especially in an unsupervised way. In this work, we have also preliminarily investigated a similar idea by designing an action-agnostic memory bank. However, as shown in Table 3, the performance is unsatisfactory. The reason could be that clustering over all the samples in an unsupervised manner could miss some important motion contexts due to the long-tailed distribution problem (refer to Q8 for more discussions). We will aim to find an appropriate way to discover such shared atomic motion dynamics (e.g., building shared memory banks across classes, considering the relations and hierarchies of classes) and will consider it in future work.
>
> **Q7: The atomic dynamics claim is the critical insight to support the whole work, but just several visualizations are shown. I suggest the authors provide more analysis to prove the atomic and common properties of the found dynamics in the memory module. And how the query works in some ideal situations, can the dynamics in memory well match the different cases.**
>
> Re: Thanks for your suggestion. In order to provide more mathematical measures on the atomic and common properties of the found dynamics ( i.e., measure whether the dynamics are representative enough), we use some common metrics to evaluate our clustering performance. The average Silhouette Coefficient for all action classes is 0.45, and the mean Variance Ratio Criterion for all action classes is 100. These metrics prove that our clustering results (i.e., motion dynamics discovery) are satisfactory. We also observe that most clustering centers (i.e., motion dynamics) represent over 4% samples and several clustering centers represent more than 10% samples. These results demonstrate that the found dynamics in the memory bank are quite representative for each action. We will also provide more visualizations on the query-and-read process in the final version, e.g., visualizing the input query motion and the matched motion dynamics in the memory bank to show how our approach works.
>
> **Q8: The possible bias from the clustering. The long-tailed distribution is common in action datasets. How it affects the clustering and representative dynamics finding?**
>
> Re: When the action class distribution is long-tailed, if we utilize a class-agnostic memory bank (i.e., clustering over all the samples without considering the action labels), the classes with more samples will dominate the representative dynamics discovery. This will introduce bias and hinder the influence of classes with fewer samples. However, in this paper, we obtain motion dynamics by applying clustering for each action class, thus avoiding such bias.
>
> **Q9: Usually, many part-level actions are the essential elements in actions instead of the whole-body motions. Can finer-grained dynamics further improve the prediction?**
>
> Re: Thank you for your constructive suggestion. Since our goal is using action to guide motion prediction, we mainly follow previous works to design our approach based on the whole-body motion, which is simpler and well-studied in the community. Designing part-level memory banks to explore finer-grained dynamics is intuitive and may bring further improvement to the system. But it is more challenging since we need to ensure the consistency among the 3D motions of different body parts. Part-level/Fine-grained action modeling is not our focus and is out of the scope of this work. We would like to leave it as one of our future work.
>
> **Q10: More discussions about the difference between the two methods are essential.**
>
> Re: Our approach is significantly different from PHD, which intends to predict 3D human motion by directly formulating a regressive map between the latent motion presentations of past observed frames and future un-observed frames. PHD ignores the action context depicted in the observed video frames, which can serve as a strong prior to facilitate the prediction. In contrast, we aim to explore action context for motion prediction and design an action-specific memory bank to exploit the representative motion dynamics within each action class. Moreover, we develop a framework to explicitly utilize the information stored in the memory bank of each action class for guiding motion prediction. We will add the above discussions to the paper to emphasize the differences.
>
> **Q11: Using the pre-trained regressor from PHD. Is this fair for the comparison between the two methods? How about training from scratch? Would the improvement be marginal?**
>
> Re: Indeed, in PHD, the regressor is pre-trained on the 3D reconstruction task, and parameters of the regressor are fixed when training their framework for motion prediction, which means that the regressor does not gain any knowledge for motion prediction. We exactly follow their implementation to fix the regressor when training for motion prediction tasks. Since the architecture of our regressor is the same as PHD, using the same pre-trained parameters as that of PHD keeps a fair comparison setting than training from scratch.
>
> Meanwhile, following your suggestion, we have also trained our regressor from scratch on the Human3.6M dataset and Penn Action dataset using the same 3D reconstruction training objective as that in PHD. The performance of our 3D human mesh regressor (i.e., oracle) is slightly worse than that of PHD (PA-MPJPE 58.1mm vs. 56.9mm). The reason may be that PHD trained their model using two extra datasets (NBA [6] and InstaVariety [6]). However, for the human motion prediction task, our approach trained from scratch still outperforms PHD (see the following table). The results further demonstrate the effectiveness of our approach to explore action context for guiding our motion prediction.
>
> |                              | PA-MPJPE ↓ |      |      |      |
> | ---------------------------- | ---------: | ---: | ---: | ---: |
> | Method                       |          5 |   10 |   20 |   30 |
> | PHD                          |       61.2 | 64.4 | 67.1 | 81.1 |
> | Ours (training from scratch) |       60.7 | 62.3 | 62.9 | 76.2 |

---

> > ### Comment · Reviewer_CSBT · 2021-08-19
> > **Most concerns are addessed.**
> >
> > After reading the reviews and responses:
> >
> > - pose and shape: clearly addressed
> >
> > - v_t: conditioned but OK
> >
> > - DTW's effect: addressed
> >
> > - Fig.4: the authors said that more highlights would be provided
> >
> > - action classification: useful discussion about the possible limitation when we utilize more semantic clues. I suggest adding more discussions about this in the next version.
> >
> > - class-i: addressed.
> >
> > - multi-action simultaneously: the authors provided a way to combine all the performing actions. This sounds reasonable but may need some verifications.
> >
> > - action hierarchy: maybe adding some discussions in the limitation section would be better, which is also mentioned by reviewer 3rjL.
> >
> > - bias in clustering: basically addressed, maybe some discussion to clarify how to hand the rare actions including very few samples in the limitation section.
> >
> > - part-level: addressed, hope this can be discussed in the next version.
> >
> > - comparison with PHD: this part needs to be added to the manuscript.
> >
> > - pre-trained regressor: training from scratch decreases the performance. I think it may need more clarifications to fully compare the proposed method and its competitors, e.g., using more data to finetune the regressor to test whether we can further improve the model. And the concerns from the other reviewers like using more information (H3.6M) also need more discussions to address in the next version.
> >
> > - The AMASS results from the response are interesting.
> >
> > Overall, my main concerns are addressed, some remain. I tend to keep my rating, I'm positive about this work. After some revision, I think this work would be interesting and inspiring for the corresponding directions.

---

> ### Author Response · Authors · 2021-08-10
> **Response To Reviewer CSBT (Second Part)**
>
> **Q12: The memory only works for the pose prediction. The pose and shape are highly correlated. Can we omit this or leave it to the shared backbone or SMPL? An ablation study should be supplemented.**
>
> Re: To validate your conjecture, we conduct experiments of jointly predicting shape and pose parameters and show the results in the following table. In this implementation, we concatenate these two parameters and feed them into the prediction block. As for the memory bank, a straightforward way is storing dynamics containing both shape and pose information (i.e., concatenating shape and pose parameters as motion dynamics to build the memory bank). However, as shown in the table, this approach (termed Jointly (Pose + Shape Bank)) achieves worse performance. We attribute this to that the shape information interferes with the clustering process as body shape is unrelated to the actions or sub-processes of actions. Another alternative for the memory bank is only storing pose information (like ours) while predicting the shape and pose parameters together, and this implementation (termed Jointly (Pose Bank)) brings clear improvement. However, predicting shape parameters and pose parameters separately (our approach) achieves the best performance.
>
> |                             | PA-MPJPE ↓ |      |      |      |
> | --------------------------- | ---------: | ---: | ---: | ---: |
> | Method                      |          5 |   10 |   20 |   30 |
> | Jointly (Pose + Shape Bank) |       62.9 | 66.1 | 68.2 | 82.1 |
> | Jointly (Pose Bank)         |       60.5 | 62.9 | 64.1 | 78.6 |
> | Separately (Ours)           |       59.6 | 61.7 | 62.5 | 75.9 |
>
> **Q13: v_t: how to get the visibility indicator?**
>
> Re: Penn Action dataset provides the visibility indicator in the annotation.
>
> **Q14: w/ and w/o DTW, the performance improvements differ significantly.**
>
> Re: DTW intends to find an optimal matching between the ground truth motion sequence and predicted motion sequence by time-warping so that the similarity between them is maximized. It is used to reduce the error caused by unreliable velocity prediction. As our approach with action memory bank can better align the velocity compared to PHD, the performance improvement is relatively larger for without DTW and smaller for with DTW. Besides, we would like to point out that using DTW for evaluation is unreasonable since we have no chance of obtaining the ground truth sequences to warp the prediction in real-world applications.
>
> **Q15: For Fig. 4, maybe some indications like red boxes on the unreasonable predicted human parts would highlight the comparison more clear. The current figure is complicated for me to find the difference and how the proposed method outperforms PHD.**
>
> Re: We will provide more visual comparisons with red boxes highlighting the differences.
>
>
> Reference:
>
> [1] Will Kay, et al. The kinetics human action video dataset. arXiv, 2017
>
> [2] Chunhui Gu, et al. Ava: A video dataset of spatio-temporally localized atomic visual actions. CVPR, 2018
>
> [3] Dian Shao, et al. Finegym: A hierarchical video dataset for fine-grained action understanding. CVPR, 2020
>
> [4] Raghav Goyal, et al. The" something something" video database for learning and evaluating visual common sense. ICCV, 2017
>
> [5] Jihoon Chung, et al. Haa500: Human-centric atomic action dataset with curated videos. arXiv, 2020
>
> [6] Angjoo Kanazawa, et al. Learning 3d human dynamics from video. CVPR, 2019

---

### Official Review · Reviewer_7fUc · 2021-07-16

**Rating:** 6
**Confidence:** 4

**Summary:**

This paper proposes a system to use predicted action class and action-specific memory to guide human motion prediction. It constructs an action memory bank that consists of observed short-term and long-term human dynamics that can be later queried and used to predict future human motion. The proposed method achieves state-of-the-art human motion forecasting on the H3.6M and PennAction datasets using only videos.

**Limitations And Societal Impact:**

No, the author did not address any potential negative social impact of their work.

Human motion prediction can be used maliciously to generate videos and content based on a short clip of video input. As more and more methods can generate photorealistic human videos based on just human pose input, there is growing significance in analyzing the potential impact of such works.

**Main Review:**

Strength:

- The insight that action classes contain crucial information that can help further motion/video generation is well-motivated and intuitive. It is clear that action class could play a central role in estimating and forecasting human motion from videos.
- The idea of decomposing action dynamics into short-term and long-term memory banks is interesting and the proposed memory bank design could facilitate future research in this direction.

Weakness:

- The most straightforward way of incorporating action class into motion prediction could be using a simple one-hot vector to indicate the action class to the motion predictor.  This simple baseline seems quite an essential method to compare against and is missing from the ablation studies.
- The improvement of motion forecasting (number provided in Table 1) is unconvincing against the PHD method. While the proposed method is better in all accounts, providing a few millimeters of motion prediction accuracy boost while using more information (namely, a pre-computed memory bank that uses motion from the same dataset) is not significant enough. In the H3.6M dataset, each actor is instructed to perform the same task and has a very similar motion pattern, so having access to the same motion class (memorized) from different subjects is a non-trivial extra source of information.
- The proposed motion prediction network, memory bank, and pose regressor seem largely derivative of prior arts and lack novelty.

Minor issues:

- I am not sure why the shape parameter beta needs to be predicted per time-step, since the inherent body shape of the person in question should not change over time. Especially when both PennAction and H3.6M are single-person datasets.
- What would be more interesting to show is if the action memory bank can be used to mix and generate different future motions from the same starting video. The idea of using action to guide motion prediction is intriguing, but evaluating this task only on motion reconstruction accuracy seems like not fulfilling its full potential.
- I am curious to see how well the method can perform in clean, non-video-based Mocap datasets (e.g. AMASS[1]). The proposed method purely operates on estimated pose using PHD, while a clean MoCap dataset can be of much larger scale and better motion quality. The video aspect of the proposed work seems a bit less convincing in the light of state-of-the-art methods [2][3], often generative, that can generate future human motion based on past motions.

Writing issues:

- L31: "in specific" should be "specifically".
- L34: with different "velocities" seem out of space.

[1] Mahmood, Naureen, Nima Ghorbani, Nikolaus F. Troje, Gerard Pons-Moll, and Michael J. Black. 2019. “AMASS: Archive of Motion Capture as Surface Shapes.” Proceedings of the IEEE International Conference on Computer Vision 2019-Octob: 5441–50.

[2] Yuan, Ye, and Kris M. Kitani. 2019. “Diverse Trajectory Forecasting with Determinantal Point Processes.” arXiv, 1–14.

[3] Xu, Jingwei, Huazhe Xu, Bingbing Ni, Xiaokang Yang, Xiaolong Wang, and Trevor Darrell. 2020. “Hierarchical Style-Based Networks for Motion Synthesis.” Lecture Notes in Computer Science 12356 LNCS: 178–94.

**Time Spent Reviewing:**

5

---

> ### Author Response · Authors · 2021-08-10
> **Response To Reviewer 7fUc**
>
> **Q1: The most straightforward way of incorporating action class into motion prediction could be using a simple one-hot vector to indicate the action class to the motion predictor.**
>
> Re: Thank you for the suggestion. we have implemented the mentioned baseline and conducted experiments on the H3.6M dataset. The results are presented in the following table. As shown, encoding the action information using a one-hot vector can also improve the system performance compared to the approach without any action information. We can also observe that our approach brings much more gains compared to the one-hot-vector-based method, which demonstrates that the proposed model can better utilize action information to guide motion prediction. We will include the discussions in the main manuscript.
>
> |                           | PA-MPJPE ↓ |      |      |      |
> | ------------------------- | ---------: | ---: | ---: | ---: |
> | Method                    |          5 |   10 |   20 |   30 |
> | No action information     |       63.1 | 66.5 | 68.9 | 83.3 |
> | One-hot Vector            |       61.8 | 64.9 | 66.5 | 80.2 |
> | Action Memory Bank (Ours) |       59.6 | 61.7 | 62.5 | 75.9 |
>
> **Q2: The improvement of motion forecasting (number provided in Table 1) is unconvincing against the PHD method. While the proposed method is better in all accounts, providing a few millimeters of motion prediction accuracy boost while using more information is not significant enough.**
>
> Re: Thank you for the comment. We would like to point out that the performance gap between our approach and the oracle/upper bound is significantly decreased compared to the PHD, although the absolute value of the improvement seems not that significant. Taking the 30-frame prediction on H3.6M with DTW for example, the gap between our approach and oracle is 5.9mm, while for PHD it is 8.2mm, the relative improvement is more than 25% (absolute improvement is 2.3mm). We argue that such an improvement is considerable. In addition, when evaluating under the more realistic setting (i.e., without DTW), the absolute value of improvement compared to PHD is more significant. Moreover, the visualization results presented in Figure 4 show that our approach performs much better than PHD in terms of velocity alignment and completeness. Combining the above elements, we argue that the performance improvement is considerable and the effectiveness of the proposed approach is validated.
>
> **Q3: In the H3.6M dataset, each actor is instructed to perform the same task and has a very similar motion pattern, so having access to the same motion class (memorized) from different subjects is a non-trivial extra source of information.**
>
> Re: We agree that the same action performed by different subjects could share similar motion patterns. Nevertheless, we would like to point out that it is not a special case that only appears in the H3.6M dataset but a common phenomenon that exists in most of the actions in the real world. And it is this phenomenon that motivates us to explore such action context to improve 3D motion prediction, which has not been explicitly explored in the community before. Specifically, we construct an action memory bank to retain the common motion patterns founded in each action and explicitly guide our 3D motion prediction. Experimental results on both H3.6M and Pen Action datasets verify the effectiveness of our approach. It is worth noting that the Penn Action dataset is captured in the wild, containing larger intra-class variations than H3.6M. And our proposed method is still able to improve the 3D motion prediction in this set. Given the above facts, we argue that our system successfully extracts informative action context to boost the motion prediction accuracy.
>
> **Q4: The proposed motion prediction network, memory bank, and pose regressor seem largely derivative of prior arts and lack novelty.**
>
> Re: We acknowledge that the architecture of our motion prediction network, memory bank, and pose regressor shares some similar components with prior works [1][2][3]. However, we’d argue that our main contribution, i.e., revealing that action information is crucial for guiding motion prediction, is of great significance. We’d also like to point out that explicitly exploring action context for 3D human motion prediction/synthesis has not been studied before and it is far from straightforward as evident in Sec. 3, particularly for modeling it by action-specific memory bank with both short-term and long-term atomic motion dynamics considered. Our results validated our claim and the effectiveness of the designed framework. Combining these elements, we argue that a substantial contribution has been made and its effectiveness demonstrated.
>
> **Q5: I am not sure why the shape parameter beta needs to be predicted per time-step.**
>
> Re: Thank you for the comment. In this work, the 3D human body parameters (shape, pose, and camera parameters) are estimated from input videos with a 3D Parameter Regressor. In practice, the shape parameters could still contain some time-varying information due to appearance variation of video frames caused by occlusion, deformation, movement, and view changes, etc. With this in mind, we select to use an encoder-LSTM-decoder to predict it. Our experiments also show that predicting shape parameters per time-step benefits our motion prediction. The obtained performance is slightly better than the system with a fixed shape (e.g., average shape or the shape of last observed frame).
>
> |                | PA-MPJPE ↓ |      |      |      |
> | -------------- | ---------: | ---: | ---: | ---: |
> | Method         |          5 |   10 |   20 |   30 |
> | Copy-last      |       59.7 | 62.1 | 63.2 | 77.0 |
> | Average        |       59.7 | 62.0 | 62.9 | 76.6 |
> | Predict (Ours) |       59.6 | 61.7 | 62.5 | 75.9 |
>
> **Q6: What would be more interesting to show is if the action memory bank can be used to mix and generate different future motions from the same starting video.**
>
> Re: We agree that it is interesting to generate different future motions from the same starting video. Indeed, we have conducted some preliminary experiments by simply changing the action class label that indicates which memory bank to be queried. However, most generated results are unrecognizable and contain some unreasonable human poses. We conjecture the reason is that the model hasn't been trained with such cases (same starting video, different class labels), thus the generated results are unpredictable. Since 3D motion generation is somehow out of the scope of this work, we did not conduct an in-depth study on it. In the future, we may consider developing a model that can generate reasonable results with more careful design. For example, we can introduce attention and soft aggregation mechanisms to query several different action memory banks to generate different predictions conditioned on the same starting video.
>
> **Q7: The idea of using action to guide motion prediction is intriguing, but evaluating this task only on motion reconstruction accuracy seems like not fulfilling its full potential.**
>
> Re: To fairly compare with PHD, we simply follow the implementation of PHD and use motion reconstruction accuracy as the evaluation metric in the submission. However, other metrics such as pose variation [4], classification accuracy [4] and self distance [5] could also be used for measuring the quality of generated 3D motion.
>
> **Q8: I am curious to see how well the method can perform in clean, non-video-based Mocap datasets (e.g. AMASS[6]). The video aspect of the proposed work seems a bit less convincing in the light of state-of-the-art methods [4][5], often generative, that can generate future human motion based on past motions.**
>
> Re: In this work, we focus on the task of predicting 3D motion from input video since it is more practical in the real-world scenario and more challenging compared to the non-video-based ones. However, our approach can also be used to predict non-video Mocap data. Due to time limitation, we select to conduct experiments on the mocap data of H3.6M rather than the mentioned AMASS set, which requires some time to prepare the set (e.g., downloading and pre-processing). The results are presented in the following table. As shown, our proposed method still surpasses PHD for the prediction of clean data.
>
> |        | PA-MPJPE ↓ |      |      |      |
> | ------ | ---------: | ---: | ---: | ---: |
> | Method |          5 |   10 |   20 |   30 |
> | PHD    |       25.2 | 29.2 | 34.6 | 49.8 |
> | Ours   |       23.8 | 26.7 | 30.2 | 44.1 |
>
> **Q9: Writing issues.**
>
> Re: Thank you for pointing out them. We will correct them.
>
> **Q10: The author did not address any potential negative social impact of their work. There is growing significance in analyzing the potential impact of human motion prediction works.**
>
> Re: In this work, we mainly focus on addressing the relatively fundamental research problem, i.e., future motion prediction. It could be used for both good and malicious purposes. For example, it could be used in autonomous driving to avoid some accidents. We would also acknowledge that it could be used to generate malicious fake videos. These videos could be unfavorable to some people, damage their reputation, and cause public social problems. We will include the above discussions about the possible negative impacts of our approach.
>
>
> Reference:
>
> [1] Jason Y Zhang, et al. Predicting 3d human dynamics from video. ICCV, 2019.
>
> [2] Seoung Wug Oh, et al. Video object segmentation using space-time memory networks. In ICCV, 2019.
>
> [3] Sainbayar Sukhbaatar, et al. End-to-end memory networks. NIPS, 2015.
>
> [4] Xu, Jingwei, et al. “Hierarchical Style-Based Networks for Motion Synthesis.” ECCV, 2020.
>
> [5] Yuan, Ye, et al. “Diverse Trajectory Forecasting with Determinantal Point Processes.” arXiv, 2019.
>
> [6] Mahmood, et al. “AMASS: Archive of Motion Capture as Surface Shapes.” ICCV, 2019.

---

> > ### Comment · Reviewer_7fUc · 2021-08-21
> > **After author's response**
> >
> > Thank you for your detailed response.
> >
> > I think my concerns about naive baseline (1), performance gain (Q2), motion pattern (Q3), shape (Q5) are largely addressed. I still have some issues with the novelty (Q4) of the proposed method and hope to see some proposed metrics in the answers to Q7 implemented. It is unsurprising that using action class (one-hot-vector) and stored dynamics (memory bank) can be helpful in predicting future, short-term motion, and motion reconstruction error as a metric provides limited insight to the quality of the constructed motion On the grounds that the proposed memory bank mechanism may be used for action-conditioned motion generation, pose prediction, I would like to raise my score to 6 but remain largely neutral about the proposed method.

---

### Official Review · Reviewer_HR9t · 2021-07-23

**Rating:** 8
**Confidence:** 5

**Summary:**

The authors of the paper proposed to exploit the representative motion dynamics of each action class for 3D human motion prediction. Specifically, they constructed an action-specific memory bank to store representative motion dynamics for each action category, with a query-read process to retrieve relevant motion dynamics from the memory bank, which are consistent with the action in the observed video frames and are used as a prior knowledge to guide motion prediction. The experiment results demonstrated that the proposed approach achieved and exceeded state-of-the-art performance. The ablation analysis demonstrated that each component's contribution to the performance improvements, especially the contribution by the action context captured in the memory bank and the action constraint loss to the motion prediction performance.

**Ethical Concerns:**

There is not any significant concerns on ethical issues.

**Limitations And Societal Impact:**

There is not any significant concerns on negative societal impact.

**Main Review:**

The framework proposed by the authors of this paper to capture action context and categories as motion dynamics in the memory bank, to guide the motion prediction, was new and the results demonstrated performance improvement. The paper was well organized. The analysis of the experiments were thorough. The new framework in this paper was based on the encoder-LSTM-decoder architecture, and recently there have been other models being proposed to overcome some disadvantages of this architecture, such as the transformer based methods and etc. in a few recent publications. It is going to be better if the authors can compare their method with these new methods. Also, it is better if the authors could include thoughts on how to improve the performance of long term motion prediction.

**Time Spent Reviewing:**

3

---

> ### Author Response · Authors · 2021-08-10
> **Response To Reviewer HR9t**
>
> **Q1: It is going to be better if the authors can compare their method with new methods such as the transformer based methods.**
>
> Re: Thank you for the suggestion. We have implemented our system based on the transformer [1], in which the encoder-LSTM-decoder is replaced with the encoder-transformer-decoder architecture. The detailed comparison results on the Human3.6M dataset are presented in the following table. As shown, our action memory bank also works well with the transformer-based architecture and increases the performance by more than 3.3mm in terms of PA-MPJPE. We can also observe that the transformer-based systems perform slightly worse than the LSTM-based counterparts. The possible reason is that transformers always rely on large-scale training data to achieve good performance, while there are not enough 3D samples in our experiments.
>
> |                        | PA-MPJPE ↓ |      |      |      |
> | :--------------------- | ---------: | ---: | ---: | ---: |
> | Method                 |          5 |   10 |   20 |   30 |
> | Baseline (Transformer) |       64.2 | 67.4 | 69.4 | 82.8 |
> | Ours (Transformer)     |       60.9 | 62.7 | 63.2 | 76.3 |
> | Baseline (LSTM)        |       63.1 | 66.5 | 68.9 | 83.3 |
> | Ours (LSTM)            |       59.6 | 61.7 | 62.5 | 75.9 |
>
> **Q2: It is better if the authors could include thoughts on how to improve the performance of long term motion prediction.**
>
> Re: Thank you. We could improve the system performance for long-term motion prediction by making a more intricate design for the memory bank. In our current submission, we have only employed motion dynamics of a fixed temporal length to guide our motion prediction. Intuitively, dynamics with varied temporal lengths could be explored for obtaining better long-term prediction results. Specifically, we can construct a hierarchical structure containing several memory banks, each of which retains the motion dynamics of a certain temporal length. Based on the constructed hierarchical bank, we can further develop a selective/attention model to adaptively determine an appropriate combination of the elements in the hierarchical bank for better motion prediction. We could include the above discussions in the final version.
>
>
> Reference:
>
> [1]	Ashish Vaswani, Noam Shazeer, Niki Parmar, Jakob Uszkoreit, Llion Jones, Aidan N Gomez, Łukasz Kaiser,  and Illia Polosukhin. Attention is all you need. In Advances in neural information processing systems, pages 5998–6008, 2017.

---

### Official Review · Reviewer_3rjL · 2021-07-24

**Rating:** 7
**Confidence:** 4

**Summary:**

The paper introduces an external memory component to the baseline method PHD on the task of human action prediction. This external memory component is designed so that it clusters common actions across videos conditioned on the action class. A query-read mechanism is designed to train and retrieve similar actions using attention. This memory component is then used to provide information to the LSTM that predicts future parameters pertaining to action conditioned on the past. A combined loss of different components pertaining to the training of the different parameters is used. This method achieves promising results, and the action memory component is able to cluster relevant actions. An ablation study is also provided to check the contribution of the action memory component.

**Limitations And Societal Impact:**

A section is devoted to the limitations of the method, however this is present in the appendix. It might be prudent to mention some of the limitations in the main text itself.

The computational resources used for the experiments has been clearly detailed.

**Main Review:**

The paper is very clear in its content. There seems to be an original contribution on top of the baseline method PHD, that of a memory component. The relevant related work is detailed, both in the human action prediction space and in the memory component space.

The main contribution is the addition of the memory component to the typical human action prediction pipeline. The memory component is well motivated, and its details are clearly outlined. Figure 5 clearly shows the actions clustered in the memory component. An ablation is also provided to check the contribution of the memory component to the overall performance.

The total loss has relevant components that contribute to performance. Each component is described and its relevance explained.

Although the performance gains with the memory component are not too big, it is however important that a memory component can help guide the training of models for this task. This aspect has been explored to some level with reasonable experiments.

---

Update : I did not have many objections to begin with. However, I see that the authors have sufficiently responded to the other reviewers' comments. In particular, additional experiments were conducted to improve on ablation studies, which proved the benefit of the proposed method. The performance of the proposed method was not too big in the first place, however this does not diminish the merit of the method, as indicated by other reviewers as well. I am happy to increase my rating after reading the authors' responses to all the reviewers' comments.

**Time Spent Reviewing:**

4

---

> ### Author Response · Authors · 2021-08-10
> **Response To Reviewer 3rjL**
>
> **Q1: A section is devoted to the limitations of the method, however this is present in the appendix. It might be prudent to mention some of the limitations in the main text itself.**
>
> Re: Thank you for your suggestion. We will provide a new section "Limitations and Future Work" to summarize the limitations of our approach and our future work in the main text.

---

> > ### Comment · Reviewer_3rjL · 2021-08-21
> > **Increase rating after authors' response**
> >
> > I did not have many objections to begin with. However, I see that the authors have sufficiently responded to the other reviewers' comments. In particular, additional experiments were conducted to improve on ablation studies, which proved the benefit of the proposed method. The performance of the proposed method was not too big in the first place, however this does not diminish the merit of the method, as indicated by other reviewers as well.
> > I am happy to increase my rating after reading the authors' responses to all the reviewers' comments.

---

### Author Response · Authors · 2021-08-10
**Response To Official Review**

Thanks to all reviewers for the constructive comments. We have carefully addressed your concerns and provided detailed responses for each review.

---

### Decision · Program_Chairs · 2021-09-27

**Decision:**

Accept (Poster)

**Comment:**

All reviewers recommend accepting this paper.
The author response was well received and cleared up some of the minor points about the work.
This paper examines the idea of creating an action-specific memory bank to store motion dynamics different action categories.
The work explores a query-read process to retrieve information from the memory bank. The work provides a good set of experiments to demonstrate the effectiveness of the proposed approach, and asserts that they have achieved state-of-the-art performance on 3D human motion prediction.

The AC recommends accepting this paper.